# The Integration of Gold Nanoparticles with Polymerase Chain Reaction for Constructing Colorimetric Sensing Platforms for Detection of Health-Related DNA and Proteins

**DOI:** 10.3390/bios12060421

**Published:** 2022-06-16

**Authors:** Wanhe Wang, Xueliang Wang, Jingqi Liu, Chuankai Lin, Jianhua Liu, Jing Wang

**Affiliations:** 1Institute of Medical Research, Northwestern Polytechnical University, 127 West Youyi Road, Xi’an 710072, China; whwang0206@nwpu.edu.cn (W.W.); wangxl2021@mail.nwpu.edu.cn (X.W.); liujingqi715@mail.nwpu.edu.cn (J.L.); 6120200115@mail.jxust.edu.cn (C.L.); jhliu0422@mail.nwpu.edu.cn (J.L.); 2Research & Development Institute of Northwestern Polytechnical University in Shenzhen, 45 South Gaoxin Road, Shenzhen 518057, China; 3Collaborative Innovation Center of NPU, Shanghai 201100, China; 4Innovation Center NPU Chongqing, Northwestern Polytechnical University, Chongqing 400000, China

**Keywords:** PCR, AuNPs, colorimetric detection, DNA, proteins

## Abstract

Polymerase chain reaction (PCR) is the standard tool in genetic information analysis, and the desirable detection merits of PCR have been extended to disease-related protein analysis. Recently, the combination of PCR and gold nanoparticles (AuNPs) to construct colorimetric sensing platforms has received considerable attention due to its high sensitivity, visual detection, capability for on-site detection, and low cost. However, it lacks a related review to summarize and discuss the advances in this area. This perspective gives an overview of established methods based on the combination of PCR and AuNPs for the visual detection of health-related DNA and proteins. Moreover, this work also addresses the future trends and perspectives for PCR–AuNP hybrid biosensors.

## 1. Introduction

Polymerase chain reaction (PCR) is a powerful nucleic acid amplification tool with exponential amplification efficiency, which can amplify a single piece of DNA into thousands of pieces within a short period [1]. Since its invention by Mullis in 1984, PCR has radically transformed biological science and has now become one of the most popular tools in molecular diagnosis [2,3]. Traditionally, PCR products are identified by the laborious and time-consuming method of gel electrophoresis, which involves a tedious procedure with the need for skilled personnel [4]. Moreover, conventional PCR is limited to qualitative analysis.

The development of fluorescence PCR technology, including real-time PCR, has significantly expanded the scope of PCR techniques in different disciplines, offering the possibility to quantify the analytes [5,6]. This type of PCR technology is also advantageous in its sensitivity, repeatability and ease of operation [7]. However, it involves expensive reagents and instrumental readout, and its background significantly grows with a cycle number of over 35, which might give false-positive results and affect its applications [8]. Moreover, fluorescent dyes used are usually impaired by their photostability, and are also affected by the autofluorescence of biological samples [9,10]. This fuels the need for the development of new PCR product analysis methods, so many methods have been established for the quantitative detection of PCR products, including electrochemistry [11], radioactivity [12], circular dichroism (CD) spectroscopy [13], surface-enhanced resonance Raman scattering (SERRS) [14], and colorimetry [15,16]. Among these methods, colorimetric PCR attracts a huge amount of attention due to its low cost, visual detection, and capability for on-site detection [17]. In the sensing fields, the application of colorimetric PCR mainly focuses on measuring the variation in nucleic acids related to diseases. Since the analysis of nucleic acids is not sufficient for molecular diagnostics, immuno-PCR based on the antibody was developed for the sensitive detection of proteins, which shows a 100–10,000-fold increase in sensitivity compared to conventional immunoassays [18]. But this is compensated with the cost and operation protocol.

Gold nanoparticles (AuNPs) are the most well-studied nanoparticles, and their intriguing chemical and photophysical properties render them an integral part of nanoscience [19]’ this is due to their high photostability, ease of synthesis and functionalization [20], and strong light absorption and scattering properties [21]. AuNPs have been extensively investigated as a kind of signal transducer in biosensors [22]. Mounting AuNP-based assays have been reported for a range of analytes, such as metal ions, small molecules, and biomolecules [23,24,25]. In particular, in 1996, Mirkin developed oligonucleotide-modified AuNPs, also known as spherical nucleic acids (SNAs), enabling controllable self-assembly of AuNPs along with optical, electronic and structural property changes [26,27]. This significantly expands the scope of AuNP applications; they are widely used for biosensing, intracellular imaging, drug delivery, etc. [28,29,30]. However, SNA-involved colorimetric biosensors generally show low sensitivity, failing to fulfill the requirement of detection [31,32].

By virtue of PCR and AuNPs, PCR–AuNP hybrid colorimetric biosensors are a good solution to retain high sensitivity without compensating visual detection, on-site detection, and low cost. Although some important advances have been achieved in this area in recent years, there lacks a specific review for this topic. In this current review, we divide the combination of PCR and AuNPs into two parts based on their manner of interaction: the non-specific combination of AuNPs with a conventional PCR product, and the specific combination of AuNPs with conventional PCR. We will discuss the advances in the combination of PCR and AuNPs to fabricate colorimetric sensing platforms for the detection of health-related DNA and proteins. The advantages and drawbacks of these sensing methods will be described. Finally, the perspectives and challenges of this strategy to develop ideal analytical methods will be discussed. We envisage this review being helpful to researchers in the area of developing methods based on AuNPs to PCR.

## 2. Non-Specific Combination of AuNPs with Conventional PCR Product

Initially, AuNPs were integrated into PCR in a non-specific manner. This integration is mainly divided into two kinds: AuNPs interacting with reagents in a PCR solution to improve PCR performance, and AuNPs interacting with PCR products with a color change. For the first situation, Li et al. first found that AuNPs enhanced PCR amplification with respect to both yield and specificity in 2005 [33]. AuNPs were then intensively investigated to improve the performance of PCR, and other nanomaterials were also found to amplify the performance [34]. In this field, AuNPs mainly work as catalysts instead of signal indicators; now, this field has evolved to become nanoPCR [35,36,37,38]. However, the mechanism remains elusive, which is mainly explained by: AuNPs absorbing polymerase to modulate the amount of active polymerase; AuNPs absorbing primer to decrease the T_m_ of primers; and AuNPs absorbing the PCR product to speed up product association [37,39,40,41,42]. There are a few reviews on nanoPCR available in *Current Organic Chemistry* [43] and the *Journal of Nanoscience and Nanotechnology* [34] for an interested audience. Here, we are interested in the second situation in which AuNPs serve as a colorimetric probe.

AuNPs have strong electrostatic interactions with single-stranded DNA (ssDNA), while showing weak affinity against double-stranded DNA (dsDNA) [44], wherein the difference depends on their different electrostatic properties. In solution, ssDNA displays sufficient flexibility and uncoils, and its exposed bases form a favorable Van der Waals interaction with AuNPs. In contrast, dsDNA is more stable and rigid, its bases tend to be hidden, and its negatively charged phosphate backbone shows repulsion to the negative ions of the AuNPs’ surfaces. Thus, the favorable interactions of ssDNA with AuNPs stabilize naked AuNPs, thus prohibiting the aggregation of AuNPs in a salt solution. The rate of adsorption onto the AuNPs’ surface is also related to ssDNA length and reaction temperature, in which shorter sequences are easier to bind to AuNPs, while a higher temperature promotes this process. The aggregation of AuNPs with appropriate sizes (d > 3.5 nm) leads to interparticle surface-plasmon coupling, causing a visible color change from red to blue at nanomolar concentrations [45]. This provides a basis for the development of AuNP-based colorimetric biosensors.

In 2004, Rothberg and coworkers first combined PCR with AuNPs, serving as a signal reporter to detect partial genomic DNA based on the selective affinity of AuNPs to ssDNA [46]. The authors designed ssDNA probes that were complementary to the desired PCR product, and had melting temperatures lower than the primers. The PCR product dsDNA was dehybridized into ssDNA at 95 °C, then the mixture was annealed below the melting temperatures of the probes, causing the hybridization between dsDNA and ssDNA. When AuNPs were added to this solution, there was immediate salt-induced AuNP aggregation, along with a color change in the solution from pink to purple (Figure 1). In contrast, when the dsDNA was not complementary to the ssDNA probes, the probes were absorbed on the AuNPs surface to prevent aggregation (Figure 1Bb). The method could be applied to detect single nucleotide polymorphisms (SNPs) in a long-QT-syndrome clinical sample. However, to reduce the potential interference, it needs to add AuNP probes into the PCR products after PCR amplification.

As conventional PCR only produces product dsDNA, the coupling of AuNPs to PCR necessitates additional ssDNA probes and additional steps. An asymmetric polymerase chain reaction (As-PCR) is capable of generating ssDNA product, offering the possibility of directly combining AuNPs and PCR [47,48]. Deng et al. reported that As-PCR was used to amplify the targeted sequence into a large amount of amplified ssDNA, which bound to naked AuNPs, enabling colorimetric detection (Figure 2) [49]. The presence of the target enabled As-PCR to produce large ssDNA amplicons, which wrapped around AuNPs to stabilize nanoparticles against salt-induced aggregation, and the red color of the AuNPs remained unchanged (Figure 2). In contrast, the absence of the target did not produce ssDNA amplicons, inducing the aggregation of AuNPs with a red-to-blue color change in the solution. This method was applied to detect *Bacillus anthracis* in clinical samples. This work demonstrated that long genomic ssDNA (508 nt/bp) also effectively stabilized AuNPs, largely expanding its application in an ssDNA-producing sensing system. Recently, Chen et al. expanded this strategy for the colorimetric detection of *Salmonella*-spiked lettuce samples by using the long ssDNA of As-PCR to stabilize bare AuNPs [50]. In addition, this strategy was also applicable in the detection of a small molecule, in which As-PCR produced a paraquat aptamer, and the presence of paraquat consumed the aptamer, causing the aggregation of bare AuNPs [51].

Traditionally, discrimination of live and dead bacteria relies on culture-based methods, whereas these methods are time-consuming, labor intensive and have low sensitivity [52,53,54]. Although fluorescent probes have been widely explored for selectively staining live bacteria, these probes suffer from low sensitivity [54,55,56,57]. Meanwhile, DNA-based PCR also cannot distinguish between live and dead bacteria, as DNA remains detectable after bacteria death [58,59]. Xu and coworkers expanded the AuNP-involved As-PCR biosensor for the selective detection of live emetic *Bacillus cereus* [60]. As propidium monoazide (PMA) selectively intercalates DNA in dead cells with compromised membranes, it can covalently cross-link with DNA under UV irradiation, preventing DNA amplification [61]. After PMA treatment, the target DNA was amplified using As-PCR; then, the resultant ssDNA stabilized AuNPs against salt-induced aggregation, so the color of the solution remained red (Figure 3). This method shows good sensitivity against live emetic *Bacillus cereus* in milk with a detection limit of 3.4 × 10^2^ CFU/mL. This method was proven to be effective in discriminating emetic *Bacillus cereus* from the other eight strains in milk; moreover, it provides a solution for the detection of live pathogenic bacteria, overcoming the drawbacks of PCR-based assays in the discrimination between live and dead bacteria. However, the efficiency of As-PCR amplification is much lower than that of conventional PCR, and the sensitivity of these methods is limited.

Beyond health-related DNA and proteins, AuNPs and PCR were also combined for the detection of prostate cancer urinary biomarker *PCA3*, a non-coding RNA [62]. The thiol forward primer was used and the presence of *PCA3* generated the thiolated *PCA3* PCR products, which were further modified on AuNPs and stabilized AuNPs in the salt solution.

These methods (Table 1) rely on non-specific interactions between AuNPs and conventional PCR products through electrostatic and Van der Waals interactions. These interactions are easily affected by interferents that are present in the sample. Therefore, the application of these methods to complicated samples such as whole-blood samples and bacteria samples remains problematic due to the biofouling and non-specific binding caused [63,64]. This has prompted the development of the specific combination of AuNPs with PCR-based biosensors for genes and proteins.

## 3. Specific Combination of SNAs with PCR

The discovery of SNAs by Mirkin in 1996 enables the tailoring properties of AuNPs based on the practical requirements, largely extending the scope of AuNP applications [26,65,66]. In particular, SNAs enable DNA-meditated AuNP assembly with optical signal change, providing the basis of DNA-mediated AuNP aggregation for the colorimetric detection of DNA [67,68]. In theory, if the sequence of PCR products or nucleic acid reagents during the PCR process is complementary to the ssDNA on AuNPs, the analyte-initiated PCR process may change the status of SNA aggregation. This offers the possibility of developing biosensors based on PCR-coupled AuNP assembly/disassembly, while DNA-meditated AuNP assembly/disassembly is more stable and controllable than ssDNA-stabilized AuNP dispersion [69].

### 3.1. SNAs in Post-Processing of PCR Product

Initial efforts in the specific combination of SNAs with PCR were mainly performed for specific interactions between SNAs probes and PCR products (Table 2), in which the products were added into an SNA solution after PCR amplification. In 2012, Liang and coworkers combined As-PCR and SNAs to develop a colorimetric assay for a specific DNA sequence [70]. The authors designed two kinds of SNA, which bound to ssDNA products generated from As-PCR to form dsDNA, inducing the assembly of AuNPs along with a color change from red to pink (Figure 4). This method can detect as little as 10 pg of template DNA, and was also applied to detect clinical *Bacillus anthracis* samples. In this work, the ssDNA product provided a template for hybridizing two SNA probes, and the resulting AuNP assembly was loose; thus, their sensitivity and specificity are limited.

To further improve its sensing performance, Wang et al. utilized the As-PCR product as a G-quadruplex DNAzyme, which oxidized cysteine into cystine, leaving silver ions free and causing SNAs to self-assemble into triplex DNA-mediated AuNP networks (Figure 5) [71]. This method detected *Staphylococcus aureus* with a detection limit of 0.28 Pg. Although this method achieves precise control assembly of AuNPs based on As-PCR, it is relatively complicated, and probably susceptible to real samples.

Typically, with SNAs, it is difficult to recognize the dsDNA from routine PCR products, so the combination of SNAs and PCR generally relies on As-PCR [72,73]. To solve this problem, triplex DNA was used for the direct combination of SNAs and conventional PCR, in which the dsDNA product from PCR formed triplex DNA with ssDNA on AuNPs’ surfaces, thus inducing the self-assembly of AuNPs (Figure 6) [74]. The strategy was capable of detecting short dsDNA (49 bp) and long dsDNA (321 bp). The applicability of this method was further demonstrated to detect *Escherichia coli* with a detection limit of 1.0 Pg/L. This method is universal and simple. However, triplex DNA is less stable than dsDNA, [75,76] and its application in real samples remains unexplored.

The low cost and rapid identification of suspected infectious people is crucial for dealing with the recent pandemic of severe acute respiratory syndrome coronavirus 2 (SARS-CoV-2) [77,78,79]. The strategy above may provide a good solution for SARS-CoV-2 screening. Recently, Somoza’s group employed T7 exonuclease to treat RT-PCR products of the SARS-CoV-2 genome for the generation of ssDNA, which further opened cholesterol-modified hairpin AuNPs, changing the hydrophilic nanostructures of AuNPs into hydrophobic nanostructures along with the aggregation [80]. Recently, Ezati et al. also implemented two hairpin-SNAs to the conventional PCR assay for the colorimetric detection of SARS-CoV-2 RNA [81]. The sequence of an ssDNA was designed to be identical to the TaqMan probe in a real-time reverse-transcription PCR (RT-PCR) assay, and the presence of the target degraded the ssDNA during the amplification process [81]. Post-treatment of the PCR products by the hairpin-modified AuNP solution kept AuNPs dispersed. In contrast, the absence of the target caused the assembly of SNAs. Later on, the authors also applied a similar strategy for the visual detection of SARS-CoV-2 RNA, in which hairpin-modified AuNPs were replaced by ssDNA-modified AuNPs, which were assumed to be more stable in the salt solution [82].

**Table 2 biosensors-12-00421-t002:** Comparison of specific combinations of SNAs in post-processing of PCR product.

Detection Method	Strategy	Target	Detection Limit	Aggregation Time	Ref.
Colorimetric	As-PCR ssDNA product binds to SNAs.	Template DNA	10 Pg	Several mins	[70]
Colorimetric	As-PCR product as a G-quadruplex DNAzyme.	Genomic DNA	5.6 fg/μL	10 min	[71]
Colorimetric	PCR product and SNAs form triplex DNA.	Short DNA and long DNA	0.5 pM for short DNA 1.0 Pg/L for long DNA	—	[74]
Colorimetric	T7 exonuclease to treat RT-PCR products.	RNA	1 nM	15 min	[80]
Colorimetric	5′-exonuclease treat RT-PCR products.	RNA	6 copies	—	[81]

### 3.2. SNAs in PCR Amplification Process

The integration of SNAs into the PCR amplification process may obviate the post-treatment of the PCR product, enabling the rapid and real-time detection of DNA sequences (Table 3). To achieve this goal, Wang and coworkers reported a colorimetric PCR method for the detection of a specific DNA by using two kinds of primer-functionalized AuNPs (Figure 7) [83]. The forward primer was designed to be the same as partial target DNA, while the reverse primer was complementary to partial target DNA. The presence of target DNA initiated the extension of the reverse primer on AuNPs as a template that was complementary to the forward primer on AuNPs. Thus, the PCR amplification process led to DNA-mediated AuNP assembly. The assembly of AuNPs induced a redshift of the absorbance peak at 519 nm with a corresponding color change from red to pink/purple. The detection can be achieved after five cycles of PCR (about 15 min) in one pot. However, it is worth noting that the length of the target DNA should be over 20 nt (~6.8 nm) to overcome the steric hindrance caused by Taq polymerase. In addition, the applicability of this method in real samples is unclear.

The combination of AuNPs and PCR in one pot may suffer from the problem of the thermal desorption of ssDNA from the AuNPs’ surfaces, affecting the stability and colorimetric properties of SNAs [84,85]. To improve the thermal stability of SNAs, a simple silica coating method was used to synthesize silica-coated SNAs, which preserves the DNA-mediated color-change property (Figure 8) [86]. The presence of the template extended the two primers to produce dsDNA amplicons. The resultant dsDNA had no influence on silica-coated SNAs, in which the solution remained red. In contrast, the absence of the target left the primers intact after PCR, in which one of the primers formed dsDNA with ssDNA on the silica-coated SNAs surface. As the transformation of ssDNA into dsDNA on AuNPs’ surfaces induces the aggregation of SNAs in a high-salt solution [87], the formation of dsDNA induced AuNP aggregation in the salt solution, along with a redshift from around 520 to 530 nm (Figure 8). This method achieved closed-tube colorimetric PCR, avoiding potential contamination; however, one issue is the reliability of this method for a complicated DNA sample, as an actual sample generally contains unreacted primers and other interferants [88,89,90].

As mentioned before, conventional PCR only produces a dsDNA product, hampering its application in ssDNA-mediated AuNPs aggregation [91,92,93]. Ling and coworkers utilized the oxyethyleneglycol-bridged primers, which formed a hairpin structure to preserve ssDNA segments after PCR amplification (Figure 9) [94]. The addition of target DNA initiated the extension, opening the hairpin structure of the primers after the first cycle, leaving ssDNA segments. These segments hybridized with ssDNA on AuNPs, and thus, crosslinked two kinds of SNA probes, causing the aggregation of AuNPs with a color change from red to blue-purple. This method was also capable of detecting genomic DNA from mouse blood. This work provides a good strategy for the production of ssDNA segments by using the conventional PCR technique. Interestingly, Jiang and coworkers also applied conventional PCR for the colorimetric detection of *Salmonella typhimurium* based on dual aptamers [95]. The PCR amplicons in the presence of *Salmonella typhimurium* hybridized with ssDNA-modified AuNPs to form a highly stable network structure, which protected AuNPs from self-aggregation in a magnesium sulfate solution. Very recently, Song and coworkers utilized a cascade invader reaction with the assistance of the flap endonuclease (FEN1) to combine PCR and SNA probes for human papillomavirus (HPV) typing, achieving closed-tube and multiplexed detection [96]. It should be noted that one-pot detection using this strategy generally requires additional reagents.

Recently, Wang et al. used SNAs as a TaqMan-like signal probe to develop a one-pot, PCR-based biosensor for DNA and proteins (Figure 10) [97]. SNAs first bound to the template DNA, which were cleaved by Taq polymerase during the extension process after the first PCR cycle, producing bare AuNPs that aggregated in a high concentration of Mg^2+^ solution. This strategy was successfully applied to visually detect *Listeria monocytogenes*, with a detection limit of 1.0 Pg μL^−1^. Generally, PCR-based protein detection mainly depends on immuno-PCR, which uses an antibody to recognize and transfer protein information into a DNA signal [18]. However, immuno-PCR is cost-ineffective and complicated. The authors used proximity ligation to further apply this strategy to the detection of protein, which employed two aptamers to transfer protein information into the DNA signal through target-mediated dsDNA formation. This strategy was successfully applied to detect thrombin, with a detection limit of 0.57 nM. The authors also utilized a more sensitive dynamic light-scattering technique to detect bacteria DNA and thrombin, due to the size-dependent signal change in AuNPs. This method depended on the specific recognition of SNAs to template dsDNA, obviating the PCR cycle-number-dependent background. Theoretically, this strategy is universal for other DNA and proteins, but its sensing performance in other analytes and in clinical samples remains to be explored.

**Table 3 biosensors-12-00421-t003:** Comparison of specific combinations of SNAs in PCR amplification process.

Detection Method	Strategy	Target	Detection Limit	Detection Range	Ref.
Colorimetric	Using two kinds of primer-functionalized AuNPs.	DNA over 20 nt	0.1 fM	—	[83]
Colorimetric	Silica coating and closed-tube method.	DNA	10^5^ copies	—	[86]
Colorimetric	Utilized the oxyethyleneglycol-bridged primers.	Genomic DNA	4.3 fM	16 fM to 1.6 nM	[94]
Colorimetric	SNAs act as TaqMan-like signal probe.	DNA and protein	0.4 pM for DNA; 0.57 nM for protein	1.0 pM to 100 nM for DNA; 1.0 nM to 20 μM for protein	[97]
DLS	SNAs act as TaqMan-like signal probe.	DNA and protein	1.1 fM for DNA; 1.0 pM for protein	3.0 fM to 1.0 nM for DNA; 2.0 pM to 200 nM for protein	[97]

## 4. Conclusions and Perspectives

The wide use of PCR and AuNPs in the sensing field stimulates the interest in the combination of PCR and AuNPs to develop colorimetric PCR-based sensors. These hybrid methods may integrate the intriguing properties of AuNPs into the amplification power of PCR, achieving high sensitivity without sacrificing the visual detection, on-site detection, and low cost. The existing methods are mainly divided into two types: the non-specific combination of AuNPs with a conventional PCR product based on ssDNA-stabilized AuNPs in high salt solution; and the specific combination of SNAs with PCR based on DNA-mediated AuNPs, based on the principle of complementary base pairing. Although the above examples have shown that this strategy is feasible, like other PCR-based methods, these methods mainly focus on the detection of DNA sequences and bacteria-derived genes, and only a few are reported to detect disease-related proteins.

Currently, the combination of PCR and AuNPs for the colorimetric detection of proteins is mainly limited to the step of transforming protein information into DNA information. Although AuNPs have also been explored for combination with immuno-PCR, the use of antibodies complicates the sensing systems and raise the cost of the assays, limiting their applications [98,99,100]. The discovery of aptamer renders this transformation easier and more effective [101]. This has been exemplified by the work of Wang et al., [97] which uses two aptamers to transfer thrombin information into a DNA signal, obviating the use of an antibody. However, there is still a long way to go. It needs additional effort to further demonstrate that the combination of PCR and AuNPs is a viable strategy to detect protein biomarkers, such as viral proteins and cancer-related proteins.

Most current PCR–AuNP-based colorimetric methods only utilize the size-dependent color change in AuNPs. AuNPs also display size-dependent scattering properties. Moreover, the light-scattering intensity of AuNPs is positively proportional to their size, making AuNPs desirable dynamic light-scattering probes; thus, the employment of dynamic light-scattering allows AuNPs to be detected at levels as low as 10^−16^ M [102,103]. This also indicates that the combination of PCR and AuNPs enables multimodal sensing. Wang et al.’s work has verified the feasibility of the colorimetric and dynamic light-scattering detection of DNA and protein [97]. However, more efforts are needed, especially in the clinical applications of PCR–AuNP-based dynamic light-scattering biosensors. Moreover, the effect of biomolecules in clinical samples on AuNP aggregation should be investigated, as it may induce negative results.

In addition, it should be noted that some factors (such as ionic strength, pH, protein conformations, the size of AuNPs, the manner of ssDNA immobilization on AuNPs, and the immobilization density on AuNPs) may affect the change in AuNP signals, as AuNP aggregation depends on electrostatic interactions. Therefore, special attention should be paid to these factors in PCR–AuNP hybrid biosensors, especially in more complicated protein-sensing systems. Moreover, the state of AuNP aggregation may change and diminish over time, along with signal change, due to the lack of aggregate stability in solution which may reduce the surface-repelling force and make them undergo particle-size expansion. Thus, time-dependent measurement should be considered to obtain more accurate results in these sensing systems. Furthermore, most PCR–AuNP colorimetric methods are developed and tested in a laboratory setting. Further practical applications need to explore their feasibility in complicated sensing systems and their adaptability to portable devices. Although the recent advances in microfluidic chips provide a huge opportunity to apply this strategy to on-site detection, [104,105], this field remains unexplored. We envision that the incorporation of this system into a microfluidic chip will largely expand its applications.

## Figures and Tables

**Figure 1 biosensors-12-00421-f001:**
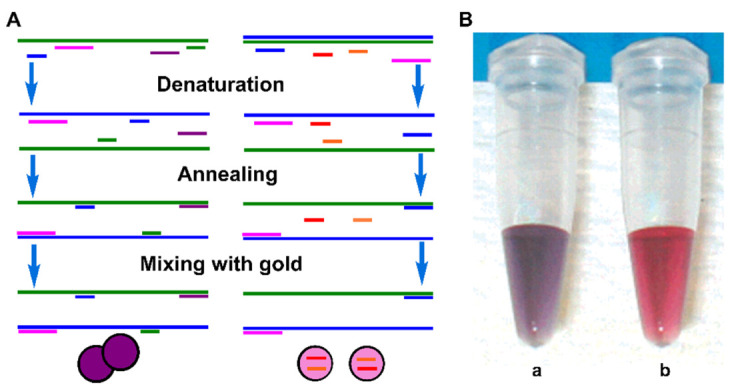
(**A**) Schematic diagram of detection of PCR product with AuNPs; (**B**) color images of the solutions with complementary ssDNA probes (**a**) and noncomplementary ssDNA probes (**b**). Adapted with permission from ref [46]. Copyright 2004 American Chemical Society.

**Figure 2 biosensors-12-00421-f002:**
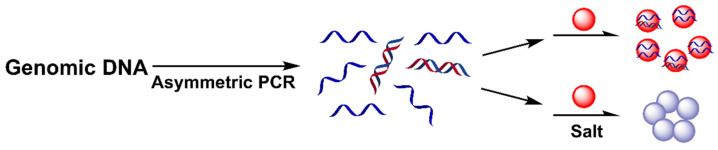
Schematic diagram of the colorimetric detection of the As-PCR DNA product with AuNP probe. Adapted with permission from ref [49]. Copyright 2012 Royal Society of Chemistry.

**Figure 3 biosensors-12-00421-f003:**
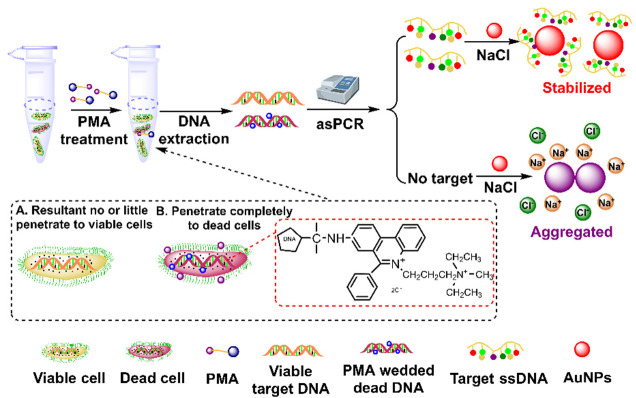
Schematic diagram of the colorimetric detection of viable emetic Bacillus cereus based on PMA-As PCR and AuNPs. Adapted with permission from ref [60]. Copyright 2018 Elsevier.

**Figure 4 biosensors-12-00421-f004:**
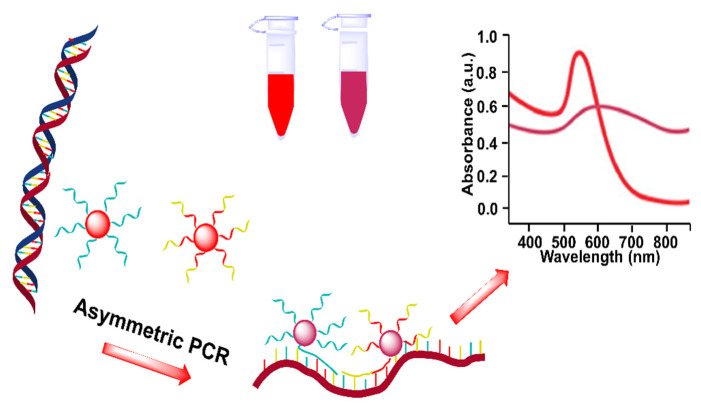
Schematic diagram of As-PCR product-assembled SNAs for colorimetric detection of *Bacillus anthracis*. Adapted with permission from ref [70]. Copyright 2012 American Chemical Society.

**Figure 5 biosensors-12-00421-f005:**
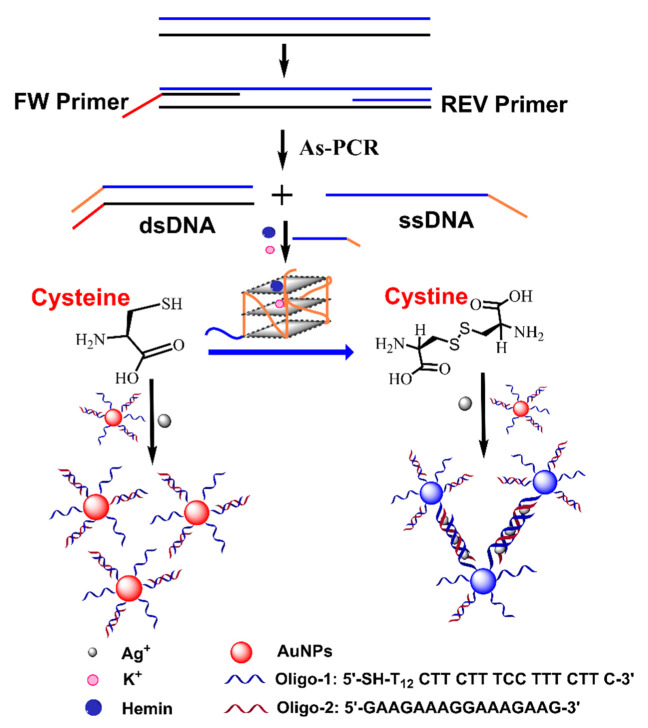
Schematic diagram of As-PCR product-derived G-quadruplex DNAzyme-mediated assembly of SNAs for colorimetric detection of *Staphylococcus aureus*. Adapted with permission from ref [71]. Copyright 2018 Springer Nature.

**Figure 6 biosensors-12-00421-f006:**
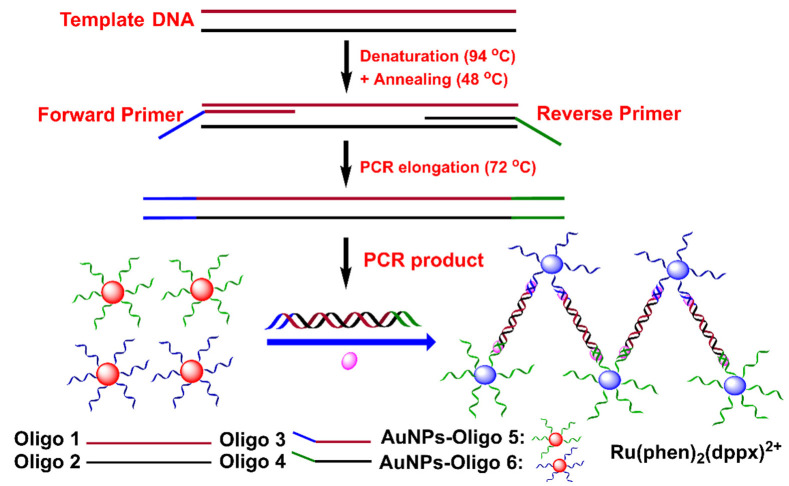
Schematic diagram of PCR product-mediated assembly of SNAs through triplex DNA formation for colorimetric detection of *Escherichia coli*. Adapted with permission from ref [74]. Copyright 2019 Elsevier.

**Figure 7 biosensors-12-00421-f007:**
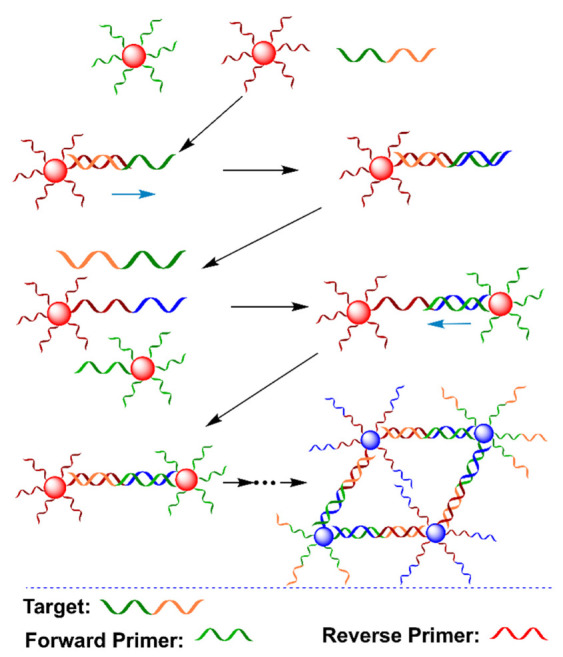
Schematic diagram of a colorimetric PCR method for a specific DNA based on primer-modified AuNP probes. Adapted with permission from ref [83]. Copyright 2010 Springer Nature.

**Figure 8 biosensors-12-00421-f008:**
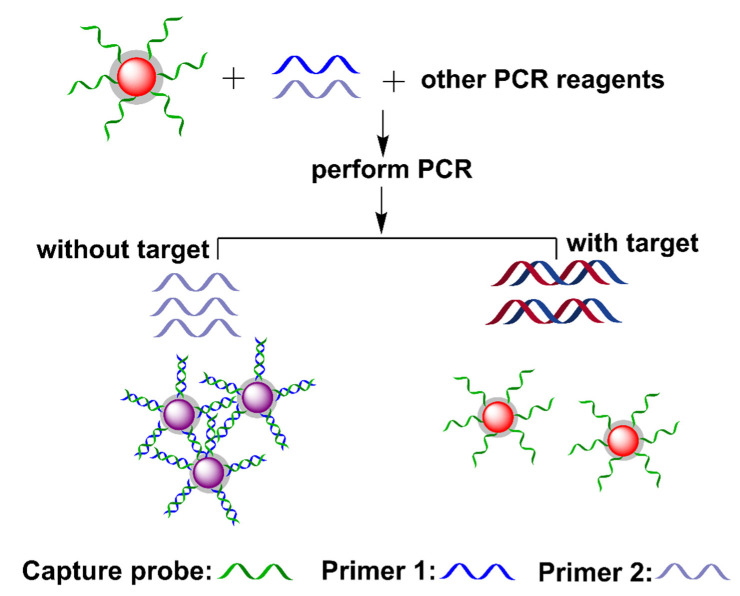
Schematic diagram of the one-step colorimetric PCR-based sensing platform. Adapted with permission from ref [86]. Copyright 2011 John Wiley and Sons.

**Figure 9 biosensors-12-00421-f009:**
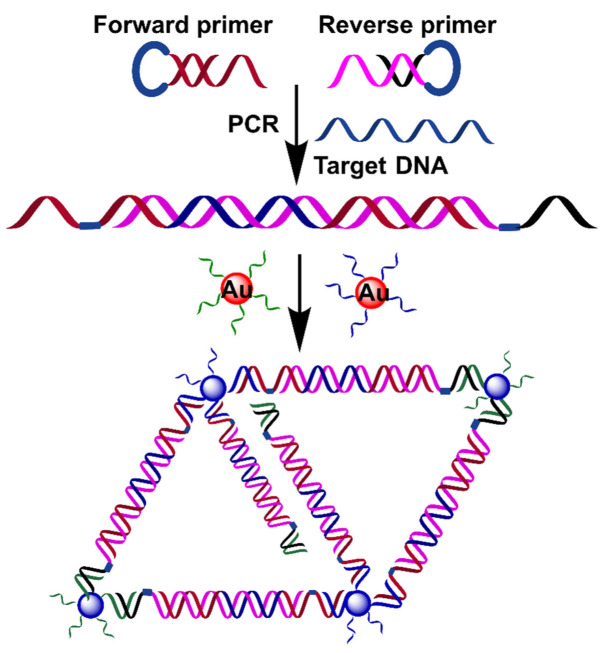
Schematic diagram of PCR dsDNA product with ssDNA-segment-induced assemblies of SNA probes for colorimetric detection of target DNA. Adapted with permission from ref [94]. Copyright 2018 Elsevier.

**Figure 10 biosensors-12-00421-f010:**
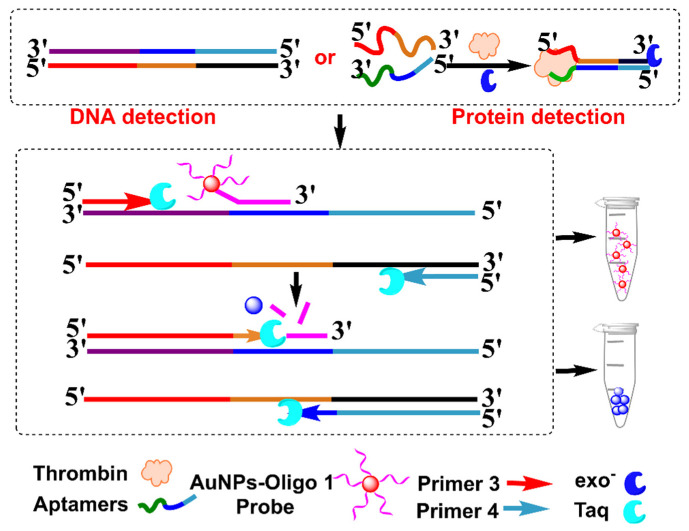
Schematic diagram of PCR-based sensing platform with SNAs as a TaqMan-like signal probe for colorimetric detection of *Listeria monocytogenes* and thrombin. Adapted with permission from ref [97]. Copyright 2019 American Chemical Society.

**Table 1 biosensors-12-00421-t001:** Comparison of non-specific combinations of AuNPs with conventional PCR product.

Detection Method	Strategy	Target	Detection Limit	Aggregation Time	Ref.
Colorimetric	ssDNA adsorbs on AuNPs without amplification.	Genomic DNA	—	Less than 1 min	[46]
Colorimetric	As-PCR ssDNA product bound to naked AuNPs.	Long genomic ssDNA	Picogram detection level	10 min	[49]
Colorimetric	PMA selectively intercalates DNA in dead cells.	Live emetic Bacillus cereus DNA	3.4 × 10^2^ CFU/mL	—	[60]
Colorimetric	Non-specific interactions between AuNPs and PCR products	Prostate cancer urinary biomarker *PCA3*	31.25 ng/reaction	—	[62]

## Data Availability

Not applicable.

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
