# Peer review of "The Integration of Gold Nanoparticles with Polymerase Chain Reaction for Constructing Colorimetric Sensing Platforms for Detection of Health-Related DNA and Proteins"

_biosensors, 2022, doi:10.3390/bios12060421_

Round 1

Reviewer 1 Report

Authors did a review on the use of gold nanoparticles as colorimetric sensors for PCRs. The manuscript is well written and easy to follow. I recommend the publication of the manuscript after minor changes. 

My only concern is about the quality of the figures. Authors should improve the figure's quality. 

Reviewer 2 Report

Since PCR was invented in 1984, it has radically transformed biological science and now became one of the most popular tools in molecular diagnosis. However, traditionally, PCR products are identified by the laborious and time-consuming gel electrophoresis, which involves a tedious procedure with the need of skilled personnel. Moreover, conventional PCR is only limited to qualitative analysis.

AuNPs are colloids with negative charges, and the color of AuNPs solution depends on the degree of AuNPs aggregation. Based on the electrostatic interactions between AuNPs and products involved in PCR inducing the color change, some methods to detect products formed in PCR could be constructed. Although some important advances have been achieved in this area in recent years, a specific review for this topic is less reported. This manuscript reviews the application of AuNPs in the detection of PCR-related products based on the specific and nonspecific interactions with PCR-related products. In my opinion, this review has positive significance for the investigation in PCR-AuNPs hybrid colorimetric biosensors. However, the following suggestions should be considered. After all, the change in AuNPs aggregation depends on electrostatic interactions, other factors (such as ionic strength, pH, and conformation of proteins, the immobilized density on AuNPs) affecting the electrostatic interactions should also be paid special attention. Therefore, in the conclusions, some recommendations should be made more clearly and not too general.

Reviewer 3 Report

1. In Section 3. Specific combination of SDAs to PCR, the authors should describe the mechanism of the specific combination clearly.

2. The authors should provide the comparisons among the methods at the end of the sections or the sub-sections.
